# Vaccine confidence and hesitancy among mothers of children under six years of age in Salvador, Brazil: The role of sociodemographic factors and health service experience

Claudia Nery Teixeira Palombo[1]*, Ednir Assis Souza[1], Érica Marvila Garcia[2], Ráren Paulo da Silva Araújo[3], Lucas Regis de Oliveira Santos[4], Marcelle Lemos Leal[5], Aline Anne Cavalcante de Oliveira[1], Ana Paula Sayuri Sato[6], Clariana Vitória Ramos de Oliveira[7]

**1** School of Nursing, Federal University of Bahia, Salvador, Bahia, Brazil, **2** Albert Einstein Israeli College of Health Sciences, São Paulo, Brazil, **3** School of Medicine of Bahia, Federal University of Bahia, Salvador, Bahia, Brazil, **4** School of Medicine, Zarns University Center, Salvador, Bahia, Brazil, **5** Municipal Health Department, Marataízes, Espírito Santo, Brazil, **6** Department of Epidemiology, School of Public Health, University of São Paulo, São Paulo, Brazil, **7** School of Nursing, University of Nevada Las Vegas, Las Vegas, Nevada, United States of America

* claudia.palombo@ufba.br

## Abstract

### Background

Vaccine hesitancy remains a pressing global health concern, particularly in early childhood, where delays or refusal to vaccinate can significantly compromise public health. Despite the recognized benefits of immunization, concerns about vaccine safety, efficacy, and necessity persist among some parents. We aimed to analyze vaccine hesitancy among mothers of children under six years old in Salvador, Bahia, Brazil.

### Method

A cross-sectional study was conducted in 2023 involving 503 mothers of children under six registered at Family Health Units in Salvador. Data were collected through structured interviews assessing socioeconomic and health characteristics, vaccination status, and attitudes toward vaccines using a standardized questionnaire. Descriptive statistics and chi-square tests were used, with a significant level of 5%.

### Results

Most mothers acknowledged the importance of vaccination, and over 80% expressed trust in vaccines; however, 27% demonstrated some degree of vaccine hesitancy. Higher maternal education (more than 11 years) was associated with greater vaccine confidence ($\beta = -0.156$, $p = 0.002$). In contrast, negative or neutral relationships with

**Data availability statement:** The data underlying this study cannot be publicly shared due to ethical and legal restrictions regarding sensitive and potentially identifying participant information. These restrictions were imposed by the Research Ethics Committee of the Federal University of Bahia (UFBA). The de-identified dataset and codebook are available to interested researchers. Requests should be directed to the Research Ethics Committee of the School of Nursing at UFBA via email at cep@ufba.br or by phone at +55 71 3283-7700.

**Funding:** National Council for Scientific and Technological Development (Universal Call Notice 2021. CNPq/MCTI/FNDCT No. 18/2021. Case: 408221/2021-6).

**Competing interests:** The authors have declared that no competing interests exist.

primary health care professionals were linked to lower trust and higher perceived vaccine risks (β = 0.123, p = 0.038). Mothers who declined new vaccines also showed significantly lower confidence in vaccination (β = 1.057, p = 0.002).

## Conclusions

Although vaccine confidence is generally high, a substantial proportion of mothers still exhibit hesitancy—often influenced by educational level, healthcare relationships, and trust in newer vaccines. These findings highlight the need for targeted strategies that strengthen provider–parent relationships and build trust in vaccine safety to reduce hesitancy and protect child health.

## Introduction

Vaccination is a cornerstone of child health, recognized globally as one of the most effective interventions for preventing infectious diseases and reducing infant mortality. It also contributes substantially to achieving the United Nations' 17 Sustainable Development Goals [1]. Despite overwhelming scientific evidence supporting vaccine efficacy and safety, vaccine acceptance has never been universal [2].

Vaccine hesitancy, the delay in acceptance or refusal of vaccines despite availability, has become a significant public health challenge, contributing to declining immunization coverage worldwide [3,4]. This phenomenon is driven by a complex interplay of cultural beliefs, political influences, misinformation, and, more recently, the intensified skepticism following the COVID-19 pandemic [5,6]. The World Health Organization has listed vaccine hesitancy among the top ten threats to global health [7]. Alarmingly, global estimates suggest that approximately 21% of parents express some degree of vaccine hesitancy, particularly regarding childhood immunizations [8].

Key concerns fueling hesitancy include doubts about vaccine efficacy and safety, especially for newly introduced vaccines, as well as complacency (low perceived risk of disease) and challenges related to access and convenience [9,10]. In recognition of the growing complexity of this issue, the WHO revised its definition of vaccine hesitancy in 2022 to incorporate social and behavioral drivers of vaccine uptake [11].

Existing literature has explored vaccine hesitancy from various perspectives, including parental decision-making [12,13], maternal attitudes [14,15], and sociodemographic determinants [16]. Studies have also addressed hesitancy among healthcare providers [17], public immunization policies [18], and the specific context of COVID-19 [19].

Given the multifaceted nature of vaccine hesitancy and its regional variations, further investigation is warranted. This study aimed to analyze vaccine hesitancy among mothers of children under six years old in Salvador, Bahia, Brazil. We hypothesize that socioeconomic characteristics and engagement with primary healthcare services are associated with maternal perceptions of vaccine confidence and disease risk.

## Method

### Study design

This was a cross-sectional study, part of a broader research project entitled 'Territorial Impact Dimensions on the Health and Nutrition Conditions of Children in Early Childhood" which aimed to understand how the physical and socioeconomic structure, service network, neighborhood, and territorial governance can affect children's health.

The study, guided by the provided guidelines Strengthening the Reporting of Observational Studies in Epidemiology [20], was conducted in primary health care units (PHCU) in Salvador, Bahia, Brazil, in 2023. The health units were selected by the Municipal Health Department.

Salvador, the capital of the State of Bahia, is in the Northeast region of Brazil. The city is very touristy, receiving people from all over the world. An estimated population of approximately 2.5 million inhabitants, making it the fourth most populous municipality in Brazil and the largest in the Northeast [21]. Salvador covers a territorial area of nearly 700 km², with an estimated population density of 3,500 inhabitants per km² [21].

The municipality is highly diverse in terms of structural dimensions and the living and working conditions of its residents. The Municipal Human Development Index stands at 0.759, which is very close to the national average of 0.760. While the urban area, tourism, and commerce are well-developed, certain neighborhoods experience significant social exclusion. In these areas, residents often rely on handicrafts, shell fishing, fishing, and informal street work for their livelihood. Additionally, racial and religious factors play a crucial role in the health-disease processes.

### Study population and sample

The study targeted mothers of children under six years old. Sample size was calculated using a 50% estimated prevalence of inappropriate feeding practices [22], based on a conservative estimate to maximize variability. A population of 199,489 children under six years of age registered in the health districts was considered, based on data from the Brazilian Institute of Geography and Statistics (IBGE) from 2010 [23], since the 2022 IBGE report had not yet been made available at the time the project was being developed, a 95% confidence level, and a 5% margin of error. The final required sample size was 388 participants.

### Inclusion and exclusion criteria

The study included children aged zero to six years who were accompanied by their biological mothers. Children with chronic or neurological conditions were excluded, as it is understood that these children require specialized care and their mothers may exhibit different behaviors regarding health care practices. When a mother accompanied more than one child within this age group, the interview focused on the youngest child.

### Data collection

Mothers and their respective children were approached at PHCU and invited to participate in the research. The interviews were conducted by trained undergraduate students from health courses, who utilized a private room to ensure the confidentiality of the participants.

Data collection was conducted from January 10 to February 28, 2023, across all health districts in Salvador, Bahia. Each interview lasted approximately 40 minutes.

The mothers were interviewed using a specific for containing variables that were categorized into two main sections: sociodemographic data of mothers (including age, race, religion, education, employment outside the home, profession or technical training, participation in a cash transfer program, and family income range) and child characteristics (which included age group, sex of the child, and health monitoring of the children). Additional data are available in Supporting Information [S1 File].

To assess the vaccine hesitancy, we applied 10-item from the Vaccine Hesitancy Scale developed by the Strategic Advisory Group of Experts Working Group (SAGE-WG/WHO) [24]. These items, which evaluate confidence, complacency, and convenience dimensions, are presented in Table 1.

## Data analysis and processing

Data were analyzed using Stata® version 15.1. Descriptive statistics were presented as means and standard deviations for continuous variables, and as frequencies and percentages for categorical variables. Maternal vaccine hesitancy was assessed using a score based on the 10-item Vaccine Hesitancy Scale [24], which was part of the survey instrument. Each item was rated on a 5-point Likert scale (1 = Strongly disagree; 2 = Disagree; 3 = Neither agree nor disagree; 4 = Agree; 5 = Strongly agree).

To ensure consistent directionality across all items, seven items (L1–L4, L6–L8) were reverse-coded, assigning higher scores to disagreement. This adjustment allowed higher overall scores to consistently reflect greater vaccine hesitancy. The final score was calculated as the mean of all item responses, representing each mother's level of hesitancy toward childhood vaccination.

Mothers were classified as vaccine hesitant if they reported delaying vaccine acceptance or refusing vaccination altogether, despite the availability of vaccination services.

To evaluate the structure of the Vaccine Hesitancy Scale, the sample was randomly divided into two subsamples. The first subsample underwent exploratory factor analysis (EFA) using Varimax rotation to identify latent factors. The second subsample was used for confirmatory factor analysis (CFA) to validate the factor structure, with items loaded exclusively onto the factors identified in the EFA [25]. Internal consistency of the scale was assessed using Cronbach's alpha.

Associations between maternal vaccine hesitancy and independent variables were examined using simple and multiple linear regression models, with the vaccine hesitancy score treated as a continuous dependent variable [26]. Variables with a p-value < 0.20 in the unadjusted analysis were included in the multivariate model using a forward selection strategy. A 5% significance level was adopted to retain variables in the final model.

Assumptions of normality and homoscedasticity were tested prior to regression modeling. Given that the distribution of the dependent variable approximated normality, parametric regression techniques were deemed appropriate. Non-parametric alternatives were considered but ultimately not required.

**Table 1. Vaccine Hesitancy Scale from the Strategic Advisory Group of Experts Working Group (SAGE-WG/WHO).**

| L1 | **Vaccines are important for my child's health** |
|----|--------------------------------------------------|
| L2 | Vaccines are effective |
| L3 | Vaccinating my child is important for the health of other children in my neighborhood |
| L4 | All childhood vaccines provided by the government are beneficial |
| L5 | New vaccines pose more risks than old ones |
| L6 | I trust the information I have received about vaccines |
| L7 | Vaccination is a good way to protect my child from diseases |
| L8 | I generally follow the vaccination guidelines recommended by my child's health care professionals |
| L9 | I worry about serious adverse events from vaccines |
| L10 | My child does not need vaccines for diseases that are no longer common |

Source: Larson et al 2015 [24].

## Quality control

Quality control measures included the use of pre-tested and standardized instruments, the preparation of a manual with detailed guidelines for conducting interviews and completing the forms, and thorough training for the entire data collection team. Additionally, a random sample of 5% of the interviews was repeated by a supervisor to verify the quality and accuracy of the information.

## Ethical considerations

This study was approved by the Research Ethics Committee at the School of Nursing of the Federal University of Bahia (CAAE: 64750722.0.0000.5531) and authorized by the Municipal Health Department of Salvador, Bahia, Brazil. Informed consent was obtained from all participating mothers. As the study involved children under six years of age, assent was not required. The research complied with the Declaration of Helsinki and Resolution 466/2012 of the National Health Council.

## Inclusivity in global research

Additional information regarding the ethical, cultural, and scientific considerations specific to inclusivity in global research is included in the Supporting Information (S2 File).

## Results

A total of 503 pairs of mothers and their children, aged zero to six years, participated in this study. Most of the mothers (78.8%) were between 20 and 40 years old, 94.2% identified as Black, 62.6% did not work outside the home, and 65.7% participated in an income transfer program. Regarding the children, the mean age was 2.8 years (SD = 1.9 years), with 50.5% being female. Notably, only 67.3% of the children were monitored at the PHCU in the neighborhood. As for vaccine hesitancy, 27% of the mothers were considered hesitant.

The distribution of responses related to the scale of vaccine hesitancy is shown in Fig 1. The shades of red represent negative behavior, the shades of gray and green indicate neutral and positive behavior, respectively. It was observed that almost the totality of mothers demonstrated positive behavior concerning the items: L1 - Vaccines are important for my child's health (99%); L2 - Vaccines are effective (97.8%); L7 - Taking vaccines is a good way to protect my son from

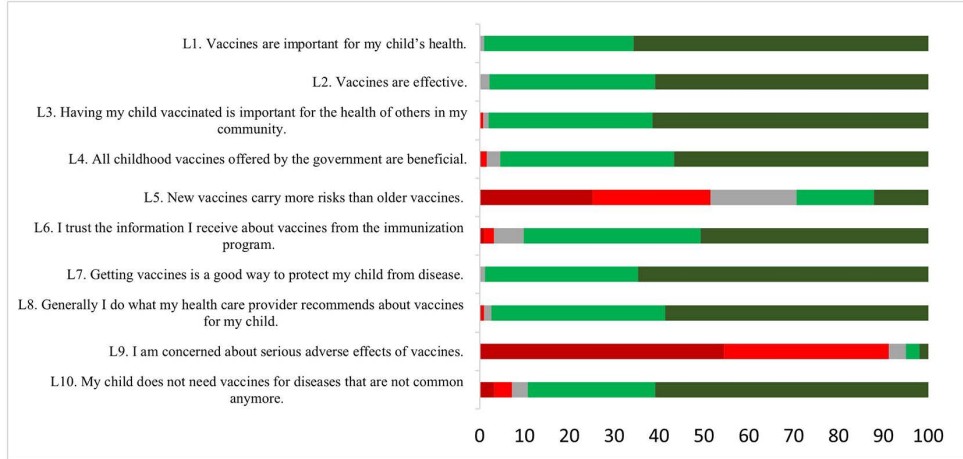

**Fig 1. Distribution of vaccine hesitancy responses in mothers of children up to six years of age according to negative behavior (dark red and red), neutral (gray), and positive behavior (dark green and green) (n = 503).** Salvador, Bahia, Brazil 2023.

diseases (98.8%). On the other hand, some mothers showed a negative attitude in relation to the items: L5 - New vaccines bring more risks than the old ones (51.4%); and L9 - I am concerned about the serious adverse effects of vaccines (91.2%). The evaluation of internal consistency revealed Cronbach's alpha of 0.8683, which indicates a good reliability of the instrument.

Table 2 shows the results of the exploratory and confirmatory factor analysis of the items on the vaccine hesitancy scale. It was found that item 10 did not correspond to any of the factors. For the other items, two factors were verified with eigenvalues of 5.09 for the lack of confidence factor and 1.15 for the risk perception factor (Table 2).

The regression models demonstrated moderate explanatory power. In the first model, the coefficient of determination ($R^2 = 0.4261$) indicates that approximately 43% of the variability in the dependent variable is accounted for by the predictors. In the second model, the $R^2$ value of 0.3279 shows that about 33% of the observed variation is explained by the independent variables. These findings suggest that both models provide a reasonable fit within the context of observational data.

Regarding socioeconomic and demographic characteristics (Table 3), the districts of Brotas (β: 0.373, p-value: < 0.001), Itapagipe (β: 0.225, p-value: 0.012), and Pau de Lima (β: 0.327, p-value: < 0.001) showed the greatest lack of confidence in vaccines. Additionally, there was a higher perception of the risk of vaccines in Barra-Rio Vermelho (β: 1.32, p-value <0.001), and Subúrbio (β: 1.034, p-value: < 0.001). Mothers with more than eleven years of schooling have greater confidence in vaccines (β: −0.156, p-value: 0.002), and those who follow Catholicism (β: 0.349, p-value: < 0.001), and the Spiritist Doctrine (β: 0.434, p-value: 0.028) have a heightened perception of vaccine risk.

No associations were found between children's health conditions and maternal lack of confidence or perceived risk regarding vaccines (Table 4).

Regarding aspects related to health services, Table 5 shows that, compared to the excellent relationship, the reasonable relationship between the mother and the health professionals of the PHCU (β: 0.123, p-value: 0.038) pointed to lower confidence in vaccines. In addition, the bad relationship indicates a greater perception of vaccine risk (Table 5).

In Table 6, the analysis showed that mothers who do not vaccinate their children with a new vaccine have lower confidence in vaccination (β: 1.057, p-value: 0.002).

## Discussion

This study aimed to analyze vaccine hesitancy among mothers of children under six years of age living in the city of Salvador, Bahia State, Brazil. The findings of this study revealed a vaccine hesitancy rate comparable to global estimates. A recent

**Table 2. Exploratory and confirmatory factor analysis of vaccine hesitancy among mothers of children under six years of age (n = 503), Salvador, Bahia, Brazil, 2023.**

| Items on the vaccine hesitancy scale | Exploratory factor analysis | | Confirmatory factor analysis | |
|---|---|---|---|---|
| | Lack of confidence | Risk perception | Lack of confidence | Risk perception |
| L1. Vaccines are important for my kid's health. | 0.8980 | −0.0943 | 0.9267 | – |
| L2. Vaccines are effective. | 0.9269 | −0.0631 | 0.8867 | – |
| L3. Vaccinating my child is important for the health of others in my community. | 0.8346 | −0.1096 | 0.8745 | – |
| L4. All childhood vaccines offered by the government are beneficial. | 0.7188 | −0.2448 | 0.8122 | – |
| L5. New vaccines carry more risks than old ones. | −0.0222 | 0.8455 | – | 0.7776 |
| L6. I trust the information I receive about vaccines from the immunization program. | 0.6146 | 0.1430 | 0.6035 | – |
| L7. Taking vaccines is a good way to protect my child from diseases. | 0.8154 | −0.0872 | 0.9319 | – |
| L8. I usually do what my doctor recommends about vaccines for my child. | 0.7652 | −0.0877 | 0.7992 | – |
| L9. I worry about the serious adverse effects of vaccines. | −0.3311 | 0.6623 | – | 0.7776 |

**Table 3. Unadjusted and adjusted linear regression models examining factors associated with lack of confidence and risk perception related to vaccine hesitancy among mothers of children under six, by socioeconomic and demographic characteristics (n = 503), Salvador, Bahia, Brazil, 2023.**

| Variables | Lack of confidence | | | | | Perception of risk | | | | |
|---|---|---|---|---|---|---|---|---|---|---|
| | Mean (SD) | Unadjusted model | | Adjusted model | | Mean (SD) | Unadjusted model | | Adjusted model | |
| | | β | p-value | β | p-value | | β | p-value | β | p-value |
| **District** | | | | | | | | | | |
| Barra – Rio Vermelho | 1.52(0.47) | Ref. | | Ref. | | 2.41(0.48) | Ref. | | Ref. | |
| Boca do Rio | 1.15(0.51) | −0.368 | <0.001 | −0.380 | <0.001 | 3.75(0.58) | 1.342 | <0.001 | 1.82 | <0.001 |
| Brotas | 1.95(0.23) | 0.435 | <0.001 | 0.373 | <0.001 | 2.20(0.62) | −0.205 | 0.233 | −0.328 | 0.154 |
| Cabula/Beiru | 1.18(0.25) | −0.337 | <0.001 | −0.347 | <0.001 | 2.77(1.01) | 0.361 | 0.011 | 0.195 | 0.313 |
| Cajazeiras | 1.19(0.34) | −0.322 | <0.001 | −0.362 | <0.001 | 3.18(0.79) | 0.772 | <0.001 | 0.364 | 0.103 |
| Centro Histórico | 1.38(0.36) | −0.139 | 0.280 | −0.144 | 0.256 | 2.64(0.50) | 0.228 | 0.366 | 0.190 | 0.562 |
| Itapagipe | 1.79(0.38) | 0.270 | 0.003 | 0.225 | 0.012 | 2.67(0.72) | 0.259 | 0.141 | −0.101 | 0.662 |
| Itapuã | 1.70(0.50) | 0.180 | 0.023 | 0.119 | 0.133 | 2.60(0.70) | 0.188 | 0.225 | 0.132 | 0.544 |
| Liberdade | 1.63(0.47) | 0.118 | 0.170 | 0.081 | 0.347 | 2.78(0.68) | 0.371 | 0.028 | 0.054 | 0.822 |
| Pau da Lima | 1.88(0.38) | 0.362 | <0.001 | 0.327 | <0.001 | 2.78(0.77) | 0.363 | 0.019 | 0.136 | 0.511 |
| São Caetano/Valéria | 1.32(0.48) | −0.193 | 0.010 | −0.244 | 0.001 | 2.84(0.92) | 0.428 | 0.004 | 0.185 | 0.380 |
| Subúrbio | 1.04(0.13) | −0.480 | <0.001 | −0.509 | <0.001 | 3.62(0.68) | 1.211 | <0.001 | 1.034 | <0.001 |
| **Age group** | | | | | | | | | | |
| <20 years old | 1.47(0.50) | Ref. | | Ref. | | 2.95(0.72) | Ref. | | | |
| 20 - 39 years old | 1.44(0.48) | −0.032 | 0.771 | – | – | 2.86(0.88) | −0.097 | 0.616 | – | – |
| >39 years old | 1.48(0.50) | 0.013 | 0.914 | – | – | 2.90(0.81) | −0.046 | 0.824 | – | – |
| **Mother's race** | | | | | | | | | | |
| Yellow | 1.43(0.46) | Ref. | | | | 2.83(0.86) | Ref. | | | |
| White | 1.48(0.52) | 0.050 | 0.271 | – | – | 2.91(0.88) | 0.074 | 0.354 | – | – |
| Indigenous | 1.37(0.47) | −0.057 | 0.602 | – | – | 2.86 (0.76) | 0.029 | 0.880 | – | – |
| Brown | 1.5(0.55) | 0.073 | 0.717 | – | – | 2.75(0.69) | −0.085 | 0.812 | – | – |
| Black | 1.43(0.0) | 0.002 | 0.997 | – | – | 3.50 (0.00) | 0.665 | 0.442 | – | – |
| **Mother's education** | | | | | | | | | | |
| <8 years | 1.54(0.53) | Ref. | | Ref. | | 3.02(0.91) | Ref. | | | |
| 8 - 11 years | 1.52(0.51) | −0.028 | 0.683 | −0.097 | 0.070 | 2.81(0.83) | −0.211 | 0.080 | – | – |
| >11 years | 1.39(0.46) | −0.156 | 0.010 | −0.156 | 0.002 | 2.85(0.86) | −0.163 | 0.129 | – | – |
| **Mother's employment** | | | | | | | | | | |
| Yes | 1.51(0.51) | Ref. | | | | 2.90(0.86) | Ref. | | | |
| No | 1.37(0.44) | −0.142 | 0.001 | – | – | 2.83(0.86) | −0.060 | 0.442 | – | – |
| **Mother's work** | | | | | | | | | | |
| Jobless | 1.35(0.42) | Ref. | | | | 2.87(0.80) | Ref. | | | |
| Formal employment | 1.51(0.47) | 0.167 | 0.022 | – | – | 2.77(0.87) | −0.102 | 0.427 | – | – |
| Informal employment | 1.45(0.51) | 0.109 | 0.082 | – | – | 2.90(0.87) | 0.030 | 0.786 | – | – |
| **Household income** | | | | | | | | | | |
| <2 SM* | 1.46(0.49) | Ref. | | | | 2.86(0.86) | Ref. | | | |
| 2 - 5 SM | 1.42(0.45) | −0.036 | 0.670 | – | – | 3.01(0.74) | 0.149 | 0.317 | – | – |
| 5 - 8 SM | 1.24(0.23) | −0.219 | 0.275 | – | – | 2.42(1.43) | −0.448 | 0.206 | – | – |
| >8 SM | 1.06(0.13 | −0.399 | 0.068 | – | – | 2.8(1.10) | −0.065 | 0.868 | – | – |
| **Cash transfer program** | | | | | | | | | | |
| Yes | 1.42(0.47) | Ref. | | | | 2.78(0.87) | Ref. | | | |
| No | 1.46(0.50) | 0.040 | 0.380 | – | – | 2.90(0.85) | 0.120 | 0.139 | – | – |

*(Continued)*

| Variables | Lack of confidence | | | | | Perception of risk | | | | |
|---|---|---|---|---|---|---|---|---|---|---|
| | Mean (SD) | Unadjusted model | | Adjusted model | | Mean (SD) | Unadjusted model | | Adjusted model | |
| | | β | p-value | β | p-value | | β | p-value | β | p-value |
| **Other children** | | | | | | | | | | |
| None | 1.67(0.58) | Ref. | | | | 3.00(0.50) | Ref. | | | |
| 1 child | 1.46(0.49) | −0.202 | 0.475 | – | – | 2.88(0.88) | −0.118 | 0.814 | – | – |
| ≥2 children | 1.43(0.49) | −0.236 | 0.406 | – | – | 2.86(0.85) | −0.142 | 0.776 | – | – |
| **Mother's religion** | | | | | | | | | | |
| None | 1.48(0.51) | Ref. | | | | 2.57(0.89) | Ref. | | | |
| Protestant | 1.41(0.46) | −0.076 | 0.603 | – | | 2.53(0.63) | −0.297 | 0.906 | −0.079 | 0.735 |
| Catholic | 1.43(0.48) | −0.053 | 0.921 | – | | 3.05(0.86) | 0.484 | <0.001 | 0.349 | <0.001 |
| Umbanda/Candomblé | 1.14(0.20) | −0.339 | 0.332 | – | | 2.00(0.00) | −0.568 | 0.350 | −0.426 | 0.430 |
| Spiritist doctrine | 1.39(0.46) | −0.093 | 0.461 | – | | 2.83(0.73) | 0.265 | 0.228 | 0.434 | 0.028 |

**Table 4. Unadjusted and adjusted linear regression models examining factors associated with lack of confidence and risk perception related to vaccine hesitancy among mothers of children under six, according to child health conditions (n = 503), Salvador, Bahia, Brazil, 2023.**

| Variables | Lack of confidence | | | | | Perception of risk | | | | |
|---|---|---|---|---|---|---|---|---|---|---|
| | Mean (SD) | Unadjusted model | | Adjusted model | | Mean (SD) | Unadjusted model | | Adjusted model | |
| | | β | p-value | β | p-value | | β | p-value | β | p-value |
| **Child health problem** | | | | | | | | | | |
| Yes | 1.44(0.48) | Ref. | | | | 2.88(0.87) | Ref. | | | |
| No | 1.59(0.53) | 0.146 | 0.117 | – | – | 2.62(0.72) | −0.262 | 0.111 | – | – |
| **Sickle cell anemia** | | | | | | | | | | |
| Yes | 1.45(0.49) | Ref. | | | | 2.87(0.86) | Ref. | | | |
| No | 1.82(0.58) | 0.376 | 0.124 | – | – | 2.75(0.87) | −0.117 | 0.788 | – | – |
| **Hospitalization** | | | | | | | | | | |
| Yes | 1.45(0.49) | Ref. | | | | 2.87(0.86) | Ref. | | | |
| No | 1.31(0.32) | −0.143 | 0.440 | – | – | 3.00(0.76) | 0.133 | 0.686 | – | – |
| **COVID-19** | | | | | | | | | | |
| Yes | 1.45(0.49) | Ref. | | | | 2.88(0.86) | Ref. | | | |
| No | 1.45(0.46) | −0.001 | 0.994 | – | – | 2.59(0.92) | −0.293 | 0.119 | – | – |

systematic review reported an average hesitancy rate of 21.1% among parents of children aged 0–6 years, with regional variations ranging from 13.3% in the Americas to 27.9% in the Eastern Mediterranean [8]. These results emphasize vaccine hesitancy as a persistent global health concern and highlight its potential contribution to declining childhood immunization rates [27,28].

In Brazil, the National Immunization Program (PNI), established in the 1970s, has been instrumental in achieving high vaccination coverage and improving public health by offering free vaccines and coordinating immunization campaigns nationwide [29]. In addition to vaccine distribution, the PNI is recognized for its extensive educational outreach through mass media channels such as television and radio. These campaigns have featured nationally known characters who emphasize the importance of vaccination and have helped foster a strong pro-vaccination culture over the decades [30].

This historical commitment to immunization may be reflected in the findings of this study, where most mothers expressed positive attitudes toward the importance and effectiveness of vaccines. Nonetheless, many also reported concerns about potential adverse effects and expressed hesitancy regarding newly introduced vaccines.

**Table 5. Unadjusted and adjusted linear regression models examining factors associated with lack of confidence and risk perception related to vaccine hesitancy among mothers of children under six, according to health service-related factors (n = 503), Salvador, Bahia, Brazil, 2023.**

| Variables | Lack of confidence | | | | | Perception of risk | | | | |
|---|---|---|---|---|---|---|---|---|---|---|
| | Mean (SD) | Unadjusted model | | Adjusted model | | Mean (SD) | Unadjusted model | | Adjusted model | |
| | | β | p-value | β | p-value | | β | p-value | β | p-value |
| Health insurance | | | | | | | | | | |
| Yes | 1.47(0.50) | Ref. | | | | 2.88(0.86) | Ref. | | | |
| No | 1.35(0.41) | −0.124 | 0.023 | – | – | 2.81(0.86) | −0.074 | 0.442 | – | – |
| Home visit in the child's first week | | | | | | | | | | |
| Yes | 1.45(0.49) | Ref. | | | | 2.88(0.87) | Ref. | | | |
| No | 1.45(0.45) | −0.004 | 0.960 | – | – | 2.72(0.77) | −0.165 | 0.173 | – | – |
| Routine home visits | | | | | | | | | | |
| Yes | 1.44(0.49) | Ref. | | | | 2.88(0.86) | Ref. | | | |
| No | 1.48(0.49) | 0.039 | 0.539 | – | – | 2.85(0.85) | −0.028 | 0.802 | – | – |
| Monitoring of the child at PHCU | | | | | | | | | | |
| Yes | 1.50(0.50) | Ref. | | | | 2.79(0.83) | Ref. | | | |
| No | 1.42(0.48) | −0.078 | 0.093 | – | – | 2.91(0.88) | 0.124 | 0.132 | – | – |
| Distance between mother's home and PHCU | | | | | | | | | | |
| Up to 500 meters (2 city blocks) | 1.50(0.49) | Ref. | | | | 2.82(0.82) | Ref. | | | |
| From 500 to 1000 meters (2–4 city blocks) | 1.36(0.48) | −0.132 | 0.011 | – | – | 3.04(0.92) | 0.214 | 0.018 | – | – |
| More than 1000 meters | 1.55(0.47) | 0.058 | 0.324 | – | – | 2.63(0.76) | −0.189 | 0.066 | – | – |
| Participation in health education groups | | | | | | | | | | |
| Yes | 1.43(0.49) | Ref. | | | | 2.89(0.87) | Ref. | | | |
| No | 1.55(0.46) | 0.115 | 0.115 | – | – | 2.73(0.75) | −0.156 | 0.224 | – | – |
| Mother's relationship with health professionals | | | | | | | | | | |
| Excellent | 1.37(0.46) | Ref. | | Ref. | | 2.77(0.94) | Ref. | | | |
| Good | 1.47(0.50) | 0.103 | 0.050 | 0.039 | 0.378 | 2.91(0.86) | 0.136 | 0.145 | 0.118 | 0.242 |
| Reasonable | 1.53(0.50) | 0.160 | 0.027 | 0.123 | 0.038 | 2.89(0.79) | 0.115 | 0.367 | 0.071 | 0.622 |
| Bad | 1.39(0.52) | 0.021 | 0.891 | 0.085 | 0.425 | 3.50(0.59) | 0.730 | 0.007 | 0.853 | 0.003 |
| Indifferent | 1.86(0.31) | 0.489 | 0.047 | 0.237 | 0.195 | 2.75(0.96) | −0.020 | 0.963 | −0.222 | 0.682 |
| Do not use the service | 1.28(0.38) | −0.090 | 0.432 | −0.039 | 0.704 | 2.62(0.59) | −0.151 | 0.455 | −0.086 | 0.689 |

**Table 6. Unadjusted and adjusted linear regression models examining factors associated with lack of confidence and risk perception related to vaccine hesitancy among mothers of children under six, according to vaccination-related factors (n = 503), Salvador, Bahia, Brazil, 2023.**

| Variables | Lack of confidence | | | | | Perception of risk | | | | |
|---|---|---|---|---|---|---|---|---|---|---|
| | Mean (SD) | Unadjusted model | | Adjusted model | | Mean (SD) | Unadjusted model | | Adjusted model | |
| | | β | p-value | β | p-value | | β | p-value | β | p-value |
| Attitude towards a new vaccine | | | | | | | | | | |
| Soon take your child to be vaccinated | 1.34(0.47) | Ref. | | Ref. | | 2.86(0.97) | Ref. | | | |
| Prefer to wait see how other people react to the vaccine | 1.54(0.48) | 0.194 | <0.001 | 0.047 | 0.070 | 2.88(0.75) | 0.018 | 0.817 | – | – |
| No vaccine | 2.5(0.51) | 1.159 | 0.001 | 1.057 | 0.002 | 2.25(0.35) | −0.612 | 0.318 | – | – |
| Deliberate delay or decision not to vaccinate | | | | | | | | | | |
| Yes | 1.39(0.47) | Ref. | | | | 2.86(0.89) | Ref. | | | |
| No | 1.56(0.51) | 0.162 | 0.001 | – | – | 2.90(0.80) | 0.043 | 0.631 | – | – |

Despite the PNI's long-standing success, Brazil has recently mirrored a global decline in vaccination coverage and increased dropout rates [31–34]. Contributing factors include the spread of misinformation [35], a diminished perception of risk from previously controlled diseases, and shortcomings in vaccine distribution and healthcare infrastructure [16,29]. Moreover, the COVID-19 pandemic further disrupted routine immunization efforts by diverting resources and public attention toward pandemic response [19,36].

In addition, it is important to consider that the concern about Events Supposedly Attributable to Vaccination or Immunization (ESAVI) directly impacts the behavior of mothers, who often choose not to vaccinate their children for fear of possible adverse effects [35,37].

The study by Brown et al. [38] highlights that younger individuals are more likely to exhibit vaccine hesitancy, particularly in decisions regarding vaccination for themselves or their children. This finding is especially relevant given the sociodemographic profile of the mothers in this study – predominantly young, self-identified black women with less than eight years of education, employed in the informal sector, with household incomes below two minimum wages, enrolled in income transfer programs, and receiving care at neighborhood Health Units. Notably, higher levels of maternal education were associated with greater confidence in vaccines. Significant differences in risk perception and vaccine confidence were also observed across health districts, which vary considerably in their socioeconomic conditions, further supporting our hypothesis.

From a socioeconomic perspective, it is important to highlight that Brazil's Bolsa Família cash transfer program includes child vaccination as one of its health conditionalities [39]. This strategy has proven effective, particularly in promoting the completion and proper use of the child health booklet [40,41], by linking financial incentives to public health objectives and helping sustain high vaccination coverage among vulnerable populations.

In terms of health service factors, the findings revealed that mothers who reported a fair to poor relationship with healthcare professionals demonstrated lower confidence in vaccines and a heightened perception of risk. Trust in healthcare providers is a critical determinant of vaccine uptake and is essential to the success of immunization programs and broader public health efforts [36].

Healthcare providers play a critical role in mitigating vaccine hesitancy, with positive provider-parent relationships strongly associated with increased trust in vaccines [14]. Moreover, key dimensions identified by WHO-SAGE—such as awareness, access, and acceptability—significantly influence vaccination behavior [15]. In this context, organizational accessibility within health services becomes essential. This includes aligning vaccination room hours with parents' work schedules, ensuring physically comfortable and ventilated waiting areas, improving geographic accessibility, and expanding outreach through extramural activities.

These actions, which extend beyond the physical boundaries of health facilities, have proven effective in fostering closer ties between health services and the community, particularly in areas of greater social vulnerability [11]. Strategies such as home visits, vaccination campaigns in daycare centers, schools, neighborhood associations, churches, street markets, and local businesses enable outreach to populations that, for various reasons, do not spontaneously access health services [42]. Moreover, the role of community health workers is essential in this process, as they serve a strategic function in identifying children with delayed vaccinations, clarifying doubts, countering misinformation, and mobilizing families [42]. The integration of these initiatives with intersectoral efforts—encompassing education, social assistance, and community leadership—can strengthen vaccine confidence, reduce logistical and cultural barriers, and contribute to the recovery of the high vaccination coverage [43] historically achieved by the National Immunization Program [44].

These findings underscore the urgent need for targeted interventions and improved communication strategies within health services to address parental concerns and reduce vaccine hesitancy. Although the influence of misinformation and fake news was not directly measured in this study, it emerged as a relevant contextual factor during data collection and analysis. Anti-vaccine movements have been known to exploit the speed and reach of digital platforms to disseminate misleading narratives, which may distort public perceptions of vaccine risks and benefits [35]. In this study, only a small

proportion of mothers explicitly cited social media as a reason for not vaccinating their children. However, the broader impact of infodemics on vaccine confidence—particularly regarding newly introduced vaccines—has been widely documented in the literature [45] and may help contextualize some of the hesitancy observed. Given that the influence of misinformation was not systematically assessed, this aspect is acknowledged as a limitation of the study and should be interpreted with caution. Future research is needed to explore the specific role of digital misinformation in shaping vaccine-related attitudes and behaviors among parents in Brazil.

The rapid spread of false information often outpaces the dissemination of accurate, science-based content from official sources such as the Ministry of Health. Combined with the overwhelming volume of information circulating online, this can create confusion and fear among the population [35,46]. Moreover, a lack of transparency from health authorities – particularly regarding regulatory processes and adverse effects – may further erode public trust and increase vaccine hesitancy. Rebuilding and maintaining public confidence require clear, transparent communication and a trustworthy relationship between communities and healthcare providers [47].

This study presents several limitations that should be considered when interpreting the findings. First, the use of face-to-face interviews may have introduced social desirability bias, as participants may have provided responses they perceived as socially acceptable rather than their true beliefs, potentially underestimating the extent of vaccine hesitancy. Second, the cross-sectional design limits the ability to infer causal relationships between sociodemographic factors, perceptions, and vaccine-related behaviors. Third, the study was conducted in a single municipality—Salvador, Bahia—which may limit the generalizability of the results to other regions with different cultural, socioeconomic, and healthcare contexts. Fourth, while the study included questions about perceptions and behaviors related to vaccination, it did not assess in depth the influence of social media use or the specific content of misinformation that mothers were exposed to. The self-reported nature of the data may introduce recall bias, particularly in questions related to vaccine schedules or adverse events. In addition, the study population was composed of individuals who actively seek and utilize public healthcare services, which may limit the generalizability of the findings. This sampling frame could generate systematic bias, as those who engage with health services may differ substantially in attitudes, access, and health-seeking behaviors compared to those who do not."

Despite these limitations, this study provides valuable baseline data for understanding vaccine attitudes among mothers in Salvador and highlights critical areas for public health intervention. To our knowledge, this is the first study to explore this theme in the region, offering locally grounded insights that can inform targeted strategies in urban settings marked by socioeconomic disparities. The use of a validated scale and face-to-face interviews enabled a nuanced understanding of maternal perceptions, even if social desirability bias may have occurred. Moreover, the study population—composed of individuals actively engaged with public healthcare services—offers a relevant perspective for strengthening existing immunization programs. These findings lay the groundwork for future longitudinal and mixed-methods research that could better capture the complexity of vaccine hesitancy and its determinants, including the role of digital misinformation and the experiences of populations not routinely reached by health services. By identifying actionable gaps and opportunities, this study contributes to the development of more inclusive, responsive, and community-based public health interventions.

## Conclusion

This study highlights the relevance of vaccine hesitancy as a global public health problem, reflecting rates similar to those observed internationally. Despite the efforts of the National Immunization Program to ensure broad vaccination coverage, the results demonstrate that concerns about adverse effects and new vaccines persist among the mothers interviewed.

The study also showed that sociodemographic aspects significantly influence vaccine hesitancy, with young women with less education showing greater distrust. In addition, factors such as misinformation, socioeconomic conditions, and failures in health infrastructure contribute to greater vaccine hesitancy, following a worrying trend observed globally.

The relationship between mothers and health professionals proved to be a decisive factor in the acceptance of vaccines, highlighting the importance of effective communication strategies and organizational accessibility in health services. The impact of fake news on immunization was identified as a significant challenge, requiring actions that promote greater transparency and combat misinformation.

Finally, this study contributes to the understanding of vaccine hesitancy in the city of Salvador, reinforcing the need for intersectoral approaches that encompass education, public policies, and targeted interventions to mitigate the factors that influence vaccine hesitancy. The findings presented here may serve as a basis for future investigations and strategies aimed at promoting childhood vaccination. In this context, public policies should prioritize the strengthening of trust between communities and health professionals, the expansion of outreach activities in vulnerable areas, and the development of communication campaigns that are culturally sensitive and evidence based. Integrating vaccination efforts with social protection programs, such as conditional cash transfers, and improving the organizational accessibility of health services—through flexible hours, community engagement, and improved infrastructure—can enhance vaccine uptake and reduce inequalities. These measures are essential to ensure that immunization remains a central pillar of public health in Brazil.

## Supporting information

**S1 File. Data collection instrument.**
(DOCX)

**S2 File. PLOS' questionnaire on inclusivity in global research.**
(DOCX)

## Author contributions

**Conceptualization:** Claudia Nery Teixeira Palombo, Ednir Assis Souza, Érica Marvila Garcia, Ráren Paulo da Silva Araújo, Clariana Vitória Ramos de Oliveira.

**Data curation:** Claudia Nery Teixeira Palombo, Ráren Paulo da Silva Araújo, Lucas Regis de Oliveira Santos, Marcelle Lemos Leal, Aline Anne Cavalcante de Oliveira.

**Formal analysis:** Claudia Nery Teixeira Palombo, Érica Marvila Garcia, Lucas Regis de Oliveira Santos, Marcelle Lemos Leal, Ana Paula Sayuri Sato, Clariana Vitória Ramos de Oliveira.

**Funding acquisition:** Claudia Nery Teixeira Palombo.

**Investigation:** Claudia Nery Teixeira Palombo, Ráren Paulo da Silva Araújo.

**Methodology:** Claudia Nery Teixeira Palombo, Érica Marvila Garcia, Marcelle Lemos Leal, Ana Paula Sayuri Sato, Clariana Vitória Ramos de Oliveira.

**Project administration:** Claudia Nery Teixeira Palombo.

**Supervision:** Claudia Nery Teixeira Palombo.

**Validation:** Ana Paula Sayuri Sato.

**Writing – original draft:** Claudia Nery Teixeira Palombo, Ednir Assis Souza, Érica Marvila Garcia, Ráren Paulo da Silva Araújo, Lucas Regis de Oliveira Santos, Marcelle Lemos Leal, Aline Anne Cavalcante de Oliveira, Ana Paula Sayuri Sato, Clariana Vitória Ramos de Oliveira.

**Writing – review & editing:** Claudia Nery Teixeira Palombo, Ednir Assis Souza, Érica Marvila Garcia, Ráren Paulo da Silva Araújo, Lucas Regis de Oliveira Santos, Marcelle Lemos Leal, Aline Anne Cavalcante de Oliveira, Ana Paula Sayuri Sato, Clariana Vitória Ramos de Oliveira.

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
