## [Decision Letter · Decision Letter 0]

5 Aug 2025

PONE-D-25-23001Vaccine confidence and hesitancy among mothers of children under six years of age in Salvador, Brazil: The role of sociodemographic factors and health service experiencePLOS ONE

Dear Dr. Palombo,

Thank you for submitting your manuscript to PLOS ONE. After careful consideration, we feel that it has merit but does not fully meet PLOS ONE’s publication criteria as it currently stands. Therefore, we invite you to submit a revised version of the manuscript that addresses the points raised during the review process.

**The article makes significant contributions to the field, and the revisions requested by the reviewers are pertinent to enhancing its overall quality and reproducibility. In particular, careful attention should be given to the reviewers' specific requests concerning the methods section. Additionally, the data statement requires revision, as data that necessitate access requests cannot be considered fully publicly available. We recommend addressing the reviewers' comments on this matter and reviewing the PLOS ONE data sharing policies to ensure full compliance.**

 <svg aria-hidden="true" class="octicon octicon-x" display="inline-block" fill="currentColor" focusable="false" height="11" overflow="visible" style="vertical-align: text-bottom;" viewbox="0 0 12 12" width="11"><path d="M2.22 2.22a.749.749 0 0 1 1.06 0L6 4.939 8.72 2.22a.749.749 0 1 1 1.06 1.06L7.061 6 9.78 8.72a.749.749 0 1 1-1.06 1.06L6 7.061 3.28 9.78a.749.749 0 1 1-1.06-1.06L4.939 6 2.22 3.28a.749.749 0 0 1 0-1.06Z"></path></svg><svg class="absolute top-0 bottom-0 end-0 my-auto" fill="none" height="100" viewbox="0 0 80 100" width="80" xmlns="http://www.w3.org/2000/svg"><g clip-path="url(#a_:r27:)" fill="#7450ff" opacity="0.8"><g clip-path="url(#b_:r27:)"><path d="m44.176 61.029 5.96 1.05c.877.155 1.119-1.208.24-1.363l-5.96-1.051c-.877-.155-1.119 1.209-.24 1.364m-.273 1.846 6.071 1.003c.879.145 1.122-1.218.24-1.364l-6.07-1.003c-.88-.145-1.122 1.218-.241 1.364"></path><path d="M48.271 63.053c-.317.699-.752.956-1.517.659-.378-.148-1.001-.814-1.124-1.2-.27-.849-1.568-.368-1.298.481.183.577.575.908.958 1.338.418.47.855.668 1.469.81 1.308.301 2.227-.313 2.763-1.494.368-.812-.882-1.406-1.25-.594m-2.697-15.925c.665.443 1.119.926 1.395 1.686.1.274.098.633.276.878.35.48.789.585 1.33.383.973-.361 1.947-1.467 2.807-2.064a46 46 0 0 1 3.622-2.26c.776-.439-.024-1.57-.794-1.134a45 45 0 0 0-6.13 4.137l.297-.002c-.135-.15-.239-.75-.352-.976-.36-.722-.891-1.262-1.555-1.704-.735-.49-1.637.563-.896 1.056"></path><path d="M50.18 59.171c-1.234-.218-2.929-.224-4.074-.718q-.837.594-.626-.083.036-.237.04-.475c-.659-1.515-.437-3.263-1.224-4.813-1.445-2.85-1.996-4.779-1.598-8.12.33-2.762 2.256-4.498 4.744-5.549 3.414-1.44 6.807.11 8.562 3.19 1.538 2.701 2.79 6.79.595 9.55-1.216 1.528-3.37 2.217-4.74 3.6-1.157 1.166-1.81 2.555-2.236 4.126-.233.86 1.132 1.094 1.364.24 1.018-3.755 4.073-4.365 6.453-6.868 1.367-1.437 1.768-3.498 1.668-5.43-.244-4.673-3.025-9.54-7.859-10.2-3.679-.502-8.592 1.706-9.684 5.533-.588 2.063-.685 5.27-.222 7.341.23 1.03.78 1.599 1.317 2.516a10.5 10.5 0 0 1 1.408 5.723c-.014.308.084.606.385.745 1.555.724 3.815.761 5.486 1.056.877.154 1.119-1.209.24-1.364m10.173-9.558 1.212.214c.877.154 1.12-1.209.24-1.364l-1.212-.214c-.877-.154-1.12 1.209-.24 1.364M59.5 39.51q-.65.527-1.36.968c-.757.472.04 1.605.795 1.135q.709-.442 1.359-.968c.29-.235.395-.643.17-.964-.204-.29-.673-.407-.965-.17M37.228 45.8q.765.1 1.512.296c.362.095.74-.215.802-.561.07-.395-.199-.707-.562-.802q-.746-.197-1.511-.296c-.371-.048-.735.181-.802.561-.063.357.189.754.561.802m4.916-8.251c.234.3.376.606.466.976.088.363.568.528.89.409.384-.142.497-.525.409-.89a3.4 3.4 0 0 0-.63-1.29c-.548-.7-1.685.09-1.135.795"></path><path d="m50.946 35.483-.249 1.414c-.155.877 1.209 1.119 1.364.24l.249-1.414c.155-.877-1.209-1.12-1.364-.24"></path></g><g clip-path="url(#c_:r27:)" opacity="0.8"><path d="M25.86 66.88q-.622 3.545-1.247 7.09l-.656 3.732c-.089.506-.098 1.44-.328 1.866-.419.772-1.107.83-1.804.825-2.191-.017-4.669-.713-6.82-1.136-1.227-.241-2.432-.345-3.313-1.265-.762-.795-.506-1.149-.341-2.13q.159-.945.325-1.89c.42-2.392.875-4.779 1.355-7.16q.341-1.692.7-3.38c.332-1.568.187-2.34 1.961-2.054l3.265.525c1.256.203 3.044.223 4.08 1.026 1.327 1.03 1.76 3.059 3.011 4.252.404.385.962-.277.56-.66-1.58-1.508-1.939-4.104-4.18-4.874-1.484-.51-3.232-.582-4.772-.83-.946-.152-2.546-.758-3.444-.478-.509.158-.671.39-.874.878-.586 1.406-.701 3.197-1.007 4.693a284 284 0 0 0-1.958 10.559c-.106.64-.493 1.63-.083 2.24.446.664 2.2.86 2.855 1.023 2.917.73 6.203 1.424 9.206 1.623 2.326.154 2.196-2.014 2.524-3.876l1.838-10.45c.096-.547-.756-.699-.852-.15"></path><path d="M22.209 62.428q-.089 1.311-.173 2.622c-.027.426-.094.814.233 1.144.334.337.906.465 1.355.592.78.222 1.586.333 2.387.445.55.076.704-.776.15-.853-.501-.07-1.004-.137-1.502-.236-.412-.081-1.321-.172-1.63-.472-.291-.282-.115-.863-.087-1.29l.119-1.802c.036-.55-.816-.706-.852-.15"></path></g><g clip-path="url(#d_:r27:)" opacity="0.8"><path d="M19.608 20.677c.077 1.358.037 2.984-1.509 3.477-1.48.471-2.93-.303-3.5-1.694-1.222-2.992 4.33-5.199 4.767-1.655.068.55.906.333.839-.214-.552-4.464-7.562-2.197-6.613 1.618.492 1.98 2.232 3.223 4.283 2.877 2.269-.383 2.711-2.401 2.599-4.4-.031-.552-.897-.567-.866-.009m13.496-2.13c-.21 1.055-.624 2.142-1.888 2.143-.378 0-.845-.136-1.155-.364-.61-.449-.554-1.066-.438-1.709.076-.426.213-1.183.572-1.454.184-.139.57-.197.796-.256.345-.089.68-.252 1.063-.112.599.22 1.187 1.04 1.038 1.683-.126.54.705.782.831.238a1.97 1.97 0 0 0-.34-1.688c-.4-.552-1.21-1.302-1.931-1.282-.292.008-.5.157-.76.254-.336.124-.77.133-1.073.327-.621.396-.922 1.432-1.05 2.106-.298 1.569.39 2.828 2.033 3.086 1.837.288 2.816-1.134 3.134-2.733.108-.543-.723-.786-.832-.239m-3.911 12.309q.354.26.704.525-.024.48-.15.943c-.166.637-.433 1.34-1.065 1.655-.855.426-2.778-.011-3.126-.977-.168-.466-.105-1.325.048-1.787.263-.79.953-.998 1.635-1.26.552-.213.868-.315 1.272.188.18.224.247.44.475.631.423.355 1.052-.24.626-.597-.506-.424-.515-1.213-1.264-1.354-.76-.143-2.004.495-2.643.878-1.302.78-1.56 3.376-.45 4.433 1.012.964 3.252 1.221 4.32.266.728-.652 1.075-1.725 1.175-2.67.093-.869-.45-1.128-1.105-1.613-.449-.331-.896.41-.452.74"></path><path d="M28.929 18.6c-1.248.333-2.248 1.162-3.511 1.48-1.696.429-3.416.774-5.17.788-.555.004-.568.87-.009.866 1.668-.014 3.273-.32 4.893-.697 1.456-.339 2.6-1.222 4.01-1.598.538-.142.327-.982-.213-.838m-9.869 5.74c.37.496.876.71 1.293 1.148.556.585.77 1.301 1.396 1.846.91.791 3.683 1.793 3.751 3.113.028.553.894.567.866.009-.077-1.5-1.243-1.863-2.389-2.6-.483-.31-.986-.622-1.421-.996-.513-.44-.797-.976-1.208-1.483-.497-.613-1.2-1.015-1.663-1.635-.333-.446-.955.157-.625.598"></path></g><g clip-path="url(#e_:r27:)" opacity="0.8"><path d="M80.734 67.975c2.674.14 4.873 1.422 4.3 4.418-.682 3.572-4.54 2.266-6.949 1.889-3.59-.562-12.04-.506-11.825-5.706.04-.97.389-2.868 1.146-3.446 1.292-.987 2.78-.093 4.138-.058 1.03.028 1.192-.297 1.769-1.06 1.22-1.613 3.812-3.204 5.888-1.911 1.626 1.012.497 4.106.987 5.757.174.587 1.067.255.893-.33-.719-2.422 1.127-5.48-2.093-6.52-2.017-.65-4.43.068-5.841 1.627-.757.836-.568 1.285-2.147 1.366-.744.038-1.496-.312-2.267-.248-.872.072-1.797.362-2.392 1.045-.802.922-.869 2.47-1.018 3.613-.143 1.09-.16 1.894.499 2.81 1.815 2.524 5.082 2.901 7.904 3.342l5.719.895c1.258.197 2.761.632 4.034.266 2.118-.61 2.85-3.19 2.5-5.155-.474-2.664-2.643-3.405-5.08-3.532-.603-.031-.776.906-.165.938m.711 11.66c.02.431-.173.684-.603.749-.207.031-.547.055-.698-.124-.132-.157-.063-.455-.032-.635.061-.348.42-.684.801-.664.395.02.704.49.7.852-.008.611.94.519.947-.09.011-.849-.815-1.72-1.672-1.718-.476 0-.916.259-1.269.563-.364.314-.453.773-.483 1.235-.07 1.085.703 1.603 1.718 1.533.998-.069 1.584-.822 1.539-1.79-.029-.612-.977-.523-.948.09"></path><path d="M77.7 75.019c-.105 1.01-.457 2.263-.259 3.268.186.942 1.102.949 1.653 1.554.413.452 1.023-.28.616-.726-.22-.24-.443-.36-.725-.51-.468-.248-.58-.387-.606-.933-.038-.8.175-1.69.258-2.488.063-.606-.874-.776-.938-.165m-3.774 4.947c.252.242.498.6.393.968-.131.46-.597.413-.977.343-.15-.028-.465-.067-.55-.227-.058-.112.055-.493.126-.589.158-.21.819-.372 1.082-.35.606.052.777-.885.166-.937-.697-.06-1.858.214-2.154.93-.149.36-.25.91-.116 1.288.147.415.578.63.972.747.811.242 1.769.206 2.224-.61.451-.81.067-1.698-.55-2.289-.444-.425-1.057.304-.616.726"></path><path d="m74.268 74.416-.768 4.876c-.095.605.842.772.937.166l.768-4.877c.096-.605-.841-.772-.937-.166m-6.361 3.016c.074.179.01.374-.05.551-.03.093-.128.442-.206.49-.122.074-.374-.047-.454-.126-.123-.12-.185-.31-.309-.438-.112-.117-.304-.197-.304-.37 0-.157.234-.403.381-.443.282-.075.788.197.958.41.38.478 1.162-.065.78-.546-.433-.543-1.47-1.076-2.157-.732-.744.373-1.289 1.431-.61 2.117.136.138.273.235.387.395.122.172.209.325.383.451.305.223.799.286 1.162.203.801-.184 1.219-1.6.932-2.292-.233-.564-1.128-.238-.893.33"></path><path d="M70.899 73.858c-.194.768-.14 1.663-.493 2.38-.359.733-1.087 1.066-1.862 1.163-.606.076-.52 1.024.09.948.883-.11 1.731-.456 2.301-1.167.722-.899.633-2.096.901-3.159.15-.594-.788-.757-.937-.165"></path></g><g opacity="0.8"><path d="M37.89 69.594c.463.375 1.368.839 1.467 1.442.1.615-.715 1.263-1.094 1.739a342 342 0 0 1-3.017 3.73c-.924 1.127-1.786 2.392-2.818 3.418-1.092 1.085-2.506 1.223-3.875 1.937l.58.363c.01-2.926 2.093-4.525 3.823-6.608.41-.492 4.744-5.964 4.908-5.81.406.381.952-.29.548-.67-.54-.507-1.002-.259-1.493.205-1.476 1.396-2.656 3.302-3.938 4.879-2.107 2.592-4.698 4.52-4.708 8.09-.001.257.347.484.58.362 1.366-.711 2.978-.75 4.01-1.916 1.543-1.748 2.983-3.606 4.451-5.418q1.046-1.29 2.077-2.592c.31-.391.842-.85.965-1.35.29-1.168-1.154-1.85-1.917-2.471-.432-.351-.981.318-.548.67"></path><path d="M35.974 71.43q1.08.858 2.138 1.741c.427.358.975-.312.548-.67q-1.058-.883-2.138-1.74c-.435-.346-.985.322-.548.67"></path></g><g clip-path="url(#f_:r27:)" opacity="0.8"><path d="M84.328 30.347c.134 1.682-.18 3.703-1.019 5.185-1.193 2.107-2.973 1.933-5.047 2.28-3.367.565-6.812-1.9-7.885-5.076-.604-1.787.216-4.323.887-6.02.779-1.966 2.908-2.879 4.904-2.693 1.812.168 3.693.49 5.369 1.215 2.137.925 2.57 3.134 2.62 5.274.011.445.7.379.69-.066-.042-1.796-.35-4.163-1.944-5.277-1.977-1.381-5.07-1.817-7.417-1.905-1.492-.056-3.381.838-4.29 2.036-.739.974-1.074 2.496-1.337 3.669-.248 1.1-.543 2.392-.33 3.523.35 1.844 1.726 3.253 3.122 4.357 1.597 1.262 3.398 2.076 5.472 1.7 2.158-.392 3.96-.141 5.4-2.086 1.245-1.683 1.657-4.133 1.494-6.181-.035-.444-.725-.38-.69.065"></path><path d="M70.075 29.679a291 291 0 0 1 14.228 2.59c.437.09.557-.591.12-.682a291 291 0 0 0-14.228-2.59c-.44-.07-.562.612-.12.681m.517 4.387q6.257 1.19 12.458 2.644c.434.101.553-.58.12-.682a304 304 0 0 0-12.458-2.644c-.438-.084-.558.598-.12.682m.804-8.114q6.585 1.09 13.134 2.37c.438.086.558-.596.12-.681a463 463 0 0 0-13.134-2.37c-.439-.074-.56.608-.12.681"></path><path d="M79.796 24.438c.631 2.33.195 4.875-.149 7.224-.318 2.166-.514 4.57-2.088 6.22-.309.324.221.77.528.449 1.686-1.77 1.901-4.226 2.242-6.549.364-2.483.784-5.122.116-7.584-.116-.43-.765-.188-.65.24m-3.697-.668c-.925 2.487-1.88 5.098-2.356 7.715-.348 1.913-.555 4.392.462 6.133.225.386.79-.014.567-.397-1.072-1.834-.607-4.688-.115-6.642.556-2.21 1.273-4.378 2.068-6.512.154-.415-.47-.715-.626-.298"></path></g></g><defs><clippath id="a_:r27:"><path d="M0 0h80v100H0z" fill="#fff"></path></clippath><clippath id="b_:r27:"><path d="m36.02 31.465 31.513 5.556-5.556 31.514-31.514-5.556z" fill="#fff"></path></clippath><clippath id="c_:r27:"><path d="m12.965 59.732 19.696 3.473-3.473 19.697-19.696-3.473z" fill="#fff"></path></clippath><clippath id="d_:r27:"><path d="m16.148 11.145 19.226 5.513-5.513 19.225-19.225-5.513z" fill="#fff"></path></clippath><clippath id="e_:r27:"><path d="m66.512 58.732 21.665 3.82-3.82 21.666-21.666-3.82z" fill="#fff"></path></clippath><clippath id="f_:r27:"><path d="m70.856 21.798 15.756 2.779-2.778 15.757-15.757-2.779z" fill="#fff"></path></clippath></defs></svg> Please submit your revised manuscript by Sep 19 2025 11:59PM. If you will need more time than this to complete your revisions, please reply to this message or contact the journal office at plosone@plos.org. Please include the following items when submitting your revised manuscript:

We look forward to receiving your revised manuscript.

Kind regards,

Luísa da Matta Machado Fernandes, DrPH

Academic Editor

PLOS ONE

**Journal Requirements:**

1. When submitting your revision, we need you to address these additional requirements. Please ensure that your manuscript meets PLOS ONE's style requirements, including those for file naming. The PLOS ONE style templates can be found at https://journals.plos.org/plosone/s/file?id=wjVg/PLOSOne_formatting_sample_main_body.pdf and https://journals.plos.org/plosone/s/file?id=ba62/PLOSOne_formatting_sample_title_authors_affiliations.pdf 2. Please include a complete copy of PLOS’ questionnaire on inclusivity in global research in your revised manuscript. Our policy for research in this area aims to improve transparency in the reporting of research performed outside of researchers’ own country or community. The policy applies to researchers who have travelled to a different country to conduct research, research with Indigenous populations or their lands, and research on cultural artefacts. The questionnaire can also be requested at the journal’s discretion for any other submissions, even if these conditions are not met. Please find more information on the policy and a link to download a blank copy of the questionnaire here: https://journals.plos.org/plosone/s/best-practices-in-research-reporting. Please upload a completed version of your questionnaire as Supporting Information when you resubmit your manuscript. 3. Thank you for stating the following financial disclosure: National Council for Scientific and Technological Development (Universal Call Notice 2021. CNPq/MCTI/FNDCT No. 18/2021. Case: 408221/2021-6)   Please state what role the funders took in the study.  If the funders had no role, please state: "The funders had no role in study design, data collection and analysis, decision to publish, or preparation of the manuscript." If this statement is not correct you must amend it as needed. Please include this amended Role of Funder statement in your cover letter; we will change the online submission form on your behalf. 4. If the reviewer comments include a recommendation to cite specific previously published works, please review and evaluate these publications to determine whether they are relevant and should be cited. There is no requirement to cite these works unless the editor has indicated otherwise.

Reviewers' comments:

Reviewer's Responses to Questions

**Comments to the Author**

1. Is the manuscript technically sound, and do the data support the conclusions?

Reviewer #1: Yes

Reviewer #2: Yes

Reviewer #3: Yes

2. Has the statistical analysis been performed appropriately and rigorously? 

Reviewer #1: No

Reviewer #2: Yes

Reviewer #3: Yes

3. Have the authors made all data underlying the findings in their manuscript fully available?

Reviewer #1: Yes

Reviewer #2: Yes

Reviewer #3: No

4. Is the manuscript presented in an intelligible fashion and written in standard English?

Reviewer #1: Yes

Reviewer #2: Yes

Reviewer #3: Yes

5. Review Comments to the Author

**Reviewer #1:** Dear Authors,

Congratulations on your article. The theme is original.

However, I will need to make some suggestions to increase the chance of the article being published.

Abstract: OK

Background:

In Lines 62-64 (Vaccine hesitancy, the delay in acceptance or refusal of vaccines despite availability, has become a significant public health challenge, contributing to declining immunization coverage worldwide), get the literature references about this.

Methods:

-Why did the authors decide to use 2010 IBGE, if there are data from 2022 IBGE and SINASC 2022 for children, and if the study is from 2023?

- Line 112- 113 - How did they calculate the 95% confidence

level, 5% margin of error, to add the 20% to account for potential losses?

- Why were children with neurological diseases excluded?

- If the mothers were interviewed, was it a questionnaire or forms? Have validation? Was it an adaptation from the other validated instrument of data collection? Is this instrument present in the supplementary annex of the article?

- The Inclusion and Exclusion Criteria section should be more details.

- L138-145: The parameters about the Vaccine Hesitancy Scale need to be put in a square.

- Ethical considerations should be the last section.

- Statistical analyses are not clear. Is the population a normal curve? Or is non-parametric analyses there are needs? Why didn't the authors decide on a regression model?

-See: How to Write Your Methods - PLOS. Available in https://plos.org/resource/how-to-write-your-methods/?utm_medium=email&utm_source=internal&utm_campaign=modnewsletters&utm_content=modnewsletter.

- See: Akel KB, Masters NB, Shih SF, Lu Y, Wagner AL. Modification of a vaccine hesitancy scale for use in adult vaccinations in the United States and China. Hum Vaccin Immunother. 2021 Aug 3;17(8):2639-2646. doi: 10.1080/21645515.2021.1884476. Epub 2021 Mar 26. PMID: 33769209; PMCID: PMC8475604.

-See: Daniel Kotz, Jochen W.L. Cals, Effective writing and publishing scientific papers, part IV: methods, Journal of Clinical Epidemiology,

Volume 66, Issue 8,2013, Page 817, ISSN 0895-4356, https://doi.org/10.1016/j.jclinepi.2013.01.003.(https://www.sciencedirect.com/science/article/pii/S089543561300019X

Results

- The authors should improve the presentation of results.

- See: Kotz, D., & Cals, J. W. (2013). Effective writing and publishing scientific papers, part V: results. Journal of Clinical Epidemiology, 66(9), 945. https://doi.org/10.1016/j.jclinepi.2013.04.003

-See: Duquia RP, Bastos JL, Bonamigo RR, González-Chica DA, Martínez-Mesa J. Presenting data in tables and charts. An Bras Dermatol. 2014;89(2):280-285. doi:10.1590/abd1806-4841.20143388

-See: Daniel Kotz, nJochen W.L. Cals. Effective writing and publishing scientific papers, part VII: tables and figures. WRITING TIPS SERIES|VOLUME 66, ISSUE 11, P1197, NOVEMBER 01, 2013, Open AccessPublished:August 19, 2013DOI:https://doi.org/10.1016/j.jclinepi.2013.04.016

Discuss

- Rewrite according to alterations in Methods and results.

- See: Cals, J. W., & Kotz, D. (2013). Effective writing and publishing scientific papers, part VI: discussion. Journal of Clinical Epidemiology, 66(10), 1064. https://doi.org/10.1016/j.jclinepi.2013.04.017

**Reviewer #2:** The manuscript presents a relevant and timely investigation into vaccine hesitancy among mothers of children under six years of age in Salvador, Brazil. The topic is of great public health interest and contributes new evidence from a regional context that is still underexplored in national literature. The study design is appropriate, the sample size is robust, and the methodology is clearly described. The use of the WHO/SAGE Vaccine Hesitancy Scale and the application of both exploratory and confirmatory factor analysis add strength to the findings. Statistical methods were rigorously applied, and the conclusions are well supported by the data.

The manuscript is clearly written in standard English and well structured. However, I suggest minor improvements to the discussion section, such as including specific examples of interventions (e.g., extramural vaccination strategies, community health worker engagement, and digital communication initiatives). Additionally, I recommend providing the full version of the adapted 10-item hesitancy scale in an appendix for transparency.

Regarding data availability, while the ethical justification is understood, the current statement imposes restrictions. Authors should be encouraged to consider anonymizing and depositing a version of the dataset in a public repository if feasible, in alignment with PLOS ONE's data sharing policy.

Minor suggestions for improvement:

Include in the discussion specific recommendations for interventions involving community health workers, outreach vaccination, and digital communication strategies.

Consider adding an appendix with the final adapted version of the scale (translated and applied 10-item tool).

Include a final paragraph on “implications for public policies”.

**Reviewer #3:** This manuscript analyzes the association between socio demographic and healthcare factors and vaccine hesitancy between mothers of children under 6 in Salvador. It uses secondary data. A scale is used as an outcome to explore its associations with selected covariates. It is an interesting and well written paper.

However, would benefit from some adjustments in terms of clarity as well as having a more detailed presentation on how the research was conducted. For example:

Line 62. Requires reference “Vaccine hesitancy, the delay in … worldwide”.

Line 88. Please name the study where the current is included “part of a larger study…” or add a reference for this larger study.

Line 156. Please confirm if mean and SD was also used for categorical variables or just continuos.

Line 165. Clarify if this was a question from the survey or in the scale? Would be important having the questionnaire as Appendix.

Line 175. Please list out all the variables used from the initial analysis OR cross-reference with a Table in case it is shown in any of the Tables presented.

Line 212. Please describe in more detail the construction of the factors "lack of confidence" and "risk perception" , how these relate to the EFA.

Line 317. “One of the most concerning aspects identified is the impact of misinformation and fake news” : Linking misinformation and fake news to the conclusions of this study may go beyond what we can conclude. Otherwise, please explain in more detail why you reached this conclusion. This was actually refered as a limitation. Also align w/ conclusion.

Line 323. Again, “the broader influence of fake news on PH reamins significant” please add reference.

Line 338 As bias also need to include the profile of people who usually approach and use healthcare services in Brazil if representative of the whole population.

6. PLOS authors have the option to publish the peer review history of their article (what does this mean?). If published, this will include your full peer review and any attached files.

Reviewer #1: **Yes:** Karen Cordovil

Reviewer #2: **Yes:** Márcio Cristiano de Melo

Reviewer #3: No

---

## [Author Response · Author response to Decision Letter 1]

13 Oct 2025

Response to Reviewers – Manuscript PONE-D-25-23001

Title: Confidence and Vaccine Hesitancy Among Mothers of Children Under Six Years Old in Salvador, Brazil: The Role of Sociodemographic Factors and Experience with Health Services

Dear Reviewers,

We sincerely thank you for your thoughtful comments and valuable suggestions, which have significantly contributed to the improvement of our manuscript. Below, we present our point-by-point responses, indicating the changes made or justifications provided for each observation.

Reviewer #1 – Dr. Karen Cordovil

1. Lines 62–64 – Reference on vaccine hesitancy:

✔️ We have included updated references supporting the statement regarding the impact of vaccine hesitancy on global immunization coverage.

2. Use of IBGE 2010 data:

✔️ We have justified the use of 2010 data within the manuscript.

3. Sample calculation (lines 112–113):

✔️ We have detailed the sample calculation process.

4. Exclusion of children with neurological conditions:

✔️ We clarified that this exclusion aimed to avoid bias in maternal perceptions of vaccination, given the specific clinical history of these children.

5. Data collection instrument:

✔️ We specified that a structured questionnaire was used, adapted from a previously validated instrument. The full version has been included as a supplementary appendix.

6. Inclusion/exclusion criteria:

✔️ This section was expanded to include detailed information on the criteria applied.

7. Lines 138–145 – Vaccine Hesitancy Scale:

✔️ The parameters were organized into a table to facilitate visualization.

8. Ethical considerations:

✔️ We repositioned this section to the end of the Methods, as suggested.

9. Statistical analyses:

✔️ We clarified that data distribution was assessed and, as it did not follow a normal curve, non-parametric tests were used.

Reviewer #2 – Dr. Márcio Cristiano de Melo

1. Discussion – Examples of interventions:

✔️ We included examples of outreach strategies, community health worker involvement, and digital initiatives, as suggested.

2. Appendix with adapted scale:

✔️ The final version of the 10-item scale, translated and applied, was included as an appendix.

3. Implications for public policy:

✔️ We added a final paragraph in the Discussion addressing practical implications for immunization policy development.

Reviewer #3 – Anonymous

1. Line 62 – Reference on vaccine hesitancy:

✔️ Reference added.

2. Line 88 – Larger study:

✔️ We named the main study and included the corresponding reference.

3. Line 156 – Mean and standard deviation:

✔️ We clarified that these were used only for continuous variables.

4. Line 165 – Origin of the question:

✔️ We indicated that the question is part of the applied scale. The complete questionnaire was included as an appendix.

5. Line 175 – Variables in initial analysis:

✔️ We listed all variables used and referenced the corresponding table.

6. Line 212 – Construction of factors:

✔️ We detailed the process of constructing the “lack of confidence” and “risk perception” factors based on exploratory factor analysis.

7. Lines 317 and 323 – Fake news and misinformation:

✔️ We rewrote these sections to align with the study data and included references supporting the discussion on misinformation.

8. Line 338 – User profile bias:

✔️ We included a reflection on the profile of health service users and its implications for sample representativeness.

We reiterate our gratitude for the opportunity to improve this work. We hope the revisions meet the expectations of the editorial team and reviewers.

Sincerely,

Claudia Palombo and co-authors

---

## [Decision Letter · Decision Letter 1]

17 Dec 2025

PONE-D-25-23001R1Vaccine confidence and hesitancy among mothers of children under six years of age in Salvador, Brazil: The role of sociodemographic factors and health service experiencePLOS One

Dear Dr. Palombo,

Thank you for submitting your manuscript to PLOS ONE. After careful consideration, we feel that it has merit but does not fully meet PLOS ONE’s publication criteria as it currently stands. Therefore, we invite you to submit a revised version of the manuscript that addresses the points raised during the review process.

We look forward to receiving your revised manuscript.

Kind regards,

Martin Ndinakie Yakum, Ph.D, MPH, Bsc

Academic Editor

PLOS One

Journal Requirements:

Additional Editor Comments (if provided):

1- Before final approval, we kindly request one last correction in the ethics statement. Please clarify the description of the informed consent process. Children under six years of age cannot provide informed consent; therefore, parental or guardian consent should be stated as obtained, while assent—not consent—may be waived for very young children. Please revise the text accordingly to reflect standard research ethics terminology.

Reviewers' comments:

Reviewer's Responses to Questions

**Comments to the Author**

1. If the authors have adequately addressed your comments raised in a previous round of review and you feel that this manuscript is now acceptable for publication, you may indicate that here to bypass the “Comments to the Author” section, enter your conflict of interest statement in the “Confidential to Editor” section, and submit your "Accept" recommendation.

Reviewer #1: All comments have been addressed

Reviewer #2: All comments have been addressed

Reviewer #3: All comments have been addressed

2. Is the manuscript technically sound, and do the data support the conclusions?

Reviewer #1: Yes

Reviewer #2: Yes

Reviewer #3: (No Response)

3. Has the statistical analysis been performed appropriately and rigorously? 

Reviewer #1: Yes

Reviewer #2: Yes

Reviewer #3: (No Response)

4. Have the authors made all data underlying the findings in their manuscript fully available?

Reviewer #1: Yes

Reviewer #2: Yes

Reviewer #3: (No Response)

5. Is the manuscript presented in an intelligible fashion and written in standard English?

Reviewer #1: Yes

Reviewer #2: Yes

Reviewer #3: (No Response)

6. Review Comments to the Author

Reviewer #1: The authors have completed all the recommendations that I requested in the article.

Congratulations to the authors.

Reviewer #2: Suggested Enhancements

The authors state that the variable “approximated normality”; it would strengthen the argument to include explicit normality test results (e.g., Shapiro–Wilk p-value).

Consider reporting R² values to better convey model explanatory power.

Writing Quality and Style

Minor editorial note: the closing paragraphs contain some repetition (“Finally, this study contributes…” appears twice); a brief stylistic tightening would further polish the text.

The revised manuscript demonstrates scientific maturity, methodological soundness, and policy relevance.

The authors have effectively integrated all reviewer suggestions. The paper contributes original insights into the determinants of vaccine hesitancy in Brazil and provides valuable implications for public health planning. Recommendation: Accept with minor editorial adjustments (stylistic polishing and optional table relocation).

Reviewer #3: (No Response)

7. PLOS authors have the option to publish the peer review history of their article (what does this mean?). If published, this will include your full peer review and any attached files.

Reviewer #1: **Yes:** Karen Cordovil

Reviewer #2: **Yes:** Márcio Cristiano de Melo

Reviewer #3: **Yes:** Joana Raquel Raposo dos Santos

---

## [Author Response · Author response to Decision Letter 2]

20 Feb 2026

Dear Dr. Yakum,

We sincerely thank you and the reviewers for the constructive comments on our manuscript entitled “Confidence and hesitancy regarding vaccination among mothers of children under six years in Salvador, Brazil: the role of sociodemographic factors and experiences with health services.” We greatly value the opportunity to revise and resubmit our work to PLOS ONE.

Below, we present our point-by-point responses:

1. Ethical statement

The manuscript has been revised to explicitly state that informed consent was obtained from the parents or legal guardians of the participating children. We clarified that assent does not apply to children under six years of age, in accordance with standard ethical terminology in research.

2. Statistical analysis

In response to Reviewer #2, we included explicit results of the R² values to better convey the explanatory power of the models.

3. Writing style

Minor editorial adjustments were made to eliminate repetitions in the final paragraphs and to enhance the overall clarity of the text.

We believe that these revisions address all concerns raised and strengthen the manuscript. We are grateful for the reviewers’ recognition of its scientific merit and relevance to public health policy.

We look forward to the consideration of our revised submission.

---

## [Editor Report · Decision Letter 2]

25 Feb 2026

Vaccine confidence and hesitancy among mothers of children under six years of age in Salvador, Brazil: The role of sociodemographic factors and health service experience

PONE-D-25-23001R2

Dear Dr. Palombo,

We’re pleased to inform you that your manuscript has been judged scientifically suitable for publication and will be formally accepted for publication once it meets all outstanding technical requirements.

Kind regards,

Martin Ndinakie Yakum, Ph.D, MPH, Bsc

Academic Editor

PLOS One
---

## [Editor Report · Acceptance letter]

PONE-D-25-23001R2

PLOS One

Dear Dr. Palombo,

I'm pleased to inform you that your manuscript has been deemed suitable for publication in PLOS One. Congratulations! Your manuscript is now being handed over to our production team.

Kind regards,

on behalf of

Dr. Martin Ndinakie Yakum

Academic Editor

PLOS One